# A Narrative Review of Problems in Learning and Practicing Palliative Care in Neurology Clinics in Japan and Proposed Solutions

**DOI:** 10.3390/brainsci12121707

**Published:** 2022-12-12

**Authors:** Takeshi Oki

**Affiliations:** 1Trinity Neurology Clinic, Petit Monde SAKURA 1-A 343-3 Jyo Sakura, Chiba 285-0815, Japan; toki@trinity-neurology.com; 2Department of Neurology, Sakura Medical Center, Toho University, 564-1 Shimoshizu, Sakura 285-8741, Japan; 3Department of Medical Education, Graduate School of Medicine, Chiba University, 1-8-1 Inohana, Chuo-ku, Chiba 260-8670, Japan

**Keywords:** neurology, palliative care, clinic, education, multidisciplinary care, health care systems

## Abstract

As the understanding of the role of palliative care in neurology increases, there is the need to ensure that these developments include not only care at home and in hospitals but also in clinics. There are no reports on palliative care from neurology clinics in Japan, and this paper considers the problems and proposed solutions for improving palliative care provided at neurology clinics in Japan. In Japan, physicians in neurology clinics are extremely busy both during and after office hours with medical treatment and the preparation of various documents and are unable to conduct case conferences. Moreover, the education system for palliative care, especially for lifelong education, is not sufficient, and multidisciplinary cooperation is difficult due to the lack of specialists and their scattered locations. To improve the care provided for patients and their families, general palliative care should be included in the health insurance system with incentives and recognition, and mandatory lifelong education should be established so that all neurologists can provide palliative care. These proposals may be appropriate for other countries as palliative care in neurology is established.

## 1. Introduction

Palliative care has three key pillars: multidisciplinary assessment and management, care through an interdisciplinary team approach, and patient- and family-centered care [1]. The scope of palliative care is expanding to include chronic progressive neurological diseases [2,3], and neurologists are now required to learn palliative care and respond to palliative care requirements [4,5]. Palliative care may be divided into generalist palliative care, which is provided by all specialists and focuses on communication, symptom management, and the facilitation of decision-making, and specialist palliative care [6] for more complex issues. General palliative care should be provided by neurologists and their associated teams and provided in clinics as well as hospitals and at home. 

The incidence rates of chronic progressive neurological diseases are increasing with the increased ageing of the Japanese population [7]. Within the Japanese health care system, patients are usually seen within out-patient clinics by neurologists; this is an obvious area for the provision of palliative care. Previous studies have shown that it is possible to provide care within the outpatient setting for people with Parkinson’s disease, and this effectively helped with physical and psychosocial aspects for these patients. In the USA, research has shown that a neuro-palliative clinic of neurologists, nurses, and social workers led to improvements in quality of life, non-motor symptom burden, motor symptom severity, and completion of advance directives for people with Parkinson’s disease (PD) [8,9]. Similarly in Czechia a Multidisciplinary Team (MDT) clinic for with amyotrophic lateral sclerosis (ALS), PD, and multiple sclerosis (MS) was found to be helpful in improving symptom burden, emotional issues, social issues, non-religious spiritual care, and quality of life, and caregivers also reported similar changes (Buzgova et al., 2020) [3]. There are no reports on palliative care from outpatient neurology clinics in Japan; this narrative review considers the problems in learning and practicing palliative care in neurology clinics in Japan and proposes possible solutions.

## 2. Current Status of Neurology Clinics in Japan

### 2.1. Number of Outpatients Treated 

According to the Ministry of Health, Labour, and Welfare’s medical statistics [10,11,12], the average daily attendance at an out-patient clinic in Japan is 41.5. There are no data on neurology clinics in Japan, but at the author’s clinic, an average of 45.6 patients are seen every day (actual figures for August 2022).

### 2.2. Number of Days of Outpatient Treatment

Although data comparing the number of neurology outpatient clinic days between hospitals and clinics in Japan are absent, the average number of neurology outpatient clinic days at three hospitals in Sakura City, Chiba is 1.5 days/week [13,14,15]. The author’s clinic is the only clinic in Sakura City, Chiba that provides outpatient neurology services for 4 days/week. The figures are similar in the neighborhood of Sakura City [16].

Physicians in neurology clinics are required to prepare various collaborative documents—including hospital referral letters, visiting nurse directives, long-term care insurance attending physician’s opinion forms, and confirmation of various reports. As physicians in neurology clinics in Japan are fully engaged in patient care during clinic hours, they often have to prepare these documents and check reports after clinic hours.

## 3. Present Position about Palliative Care Clinics

In Japan, palliative care has developed mainly in hospital wards under the insurance medical system. Palliative care for non-cancer diseases has been addressed in the fields of home medicine and geriatrics since 2000 [17]. However, with the enactment of the Basic Law on Cancer Control in 2006, the use of palliative care wards was limited to cancer and AIDS. Palliative care wards have also been available for end-stage heart failure since 2018, but at present, end-stage neurological diseases can not be treated in palliative care wards.

Within out-patient services, since 2008, outpatient palliative care management fees could be claimed, but only for cancer patients. In home care, since 2006, it has been possible to claim an additional fee for home terminal care regardless of the disease. If two or more home visits are made within 14 days of death, 10,000 yen ($72) may be charged once, but this fee is not equivalent to the comprehensive home cancer treatment fee of 16,500 yen ($120)/day, which is only for cancer. Thus, palliative care for end-stage neurological diseases may be provided at home or in hospital wards, subject to a fee for guidance and management of intractable diseases [18].

## 4. Present Position of Education in Palliative Care

Neurologists do need to receive education, both theoretical and practical, for palliative care [19,20]. In Japan, there are increasing educational opportunities at all levels, as described below.

### 4.1. Under-Graduate Training

The content of under-graduate medical education is defined in the Model Core Curriculum for Medical Education [21], presented by the Ministry of Education, Culture, Sports, Science, and Technology in Japan. It is positioned as essential practical medical skills as a part of the basic medical examination knowledge. The areas to be covered within these courses are an overview of palliative care (including palliative care team, hospice, palliative care unit, and home palliative care), an understanding of holistic suffering, physical and psychosocial distresses that occur frequently in palliative care assessment of pain, pharmacotherapy for pain relief and cancer pain treatment methods, indications and challenges of the use of opioids, and the psychology of patients and families in palliative care.

### 4.2. Post-Graduate Education

The guidelines for teaching residents in Japan [22] state that ‘During training in internal medicine, surgery, and palliative care, the trainee is responsible for patients requiring palliative care and participates in the activities of the palliative care team’. In addition, they attend workshops where they can learn about palliative care systematically.

### 4.3. Specialty Training

The palliative medicine specialist program has been in place since 2010 in Japan. A palliative medicine specialist is a doctor who has undergone training for at least two years at a training facility accredited by the Japanese Society of Palliative Medicine under the guidance of a Japanese Society of Palliative Medicine certified specialist, in accordance with the training curriculum of the Japanese Society of Palliative Medicine. There are 22 training items including palliative care for neurological diseases, and students must pass a specialist examination [23]. As of 2022, 303 specialists of the Japanese Society of Palliative Medicine have been certified [24]. The Japanese Society for Palliative Medicine has a registration system for palliative care teams, and the number of registered teams in 2020 was 528 teams nationwide [25].

### 4.4. Continuing Education

The Japan Medical Association has an e-learning course on ‘Care at the end of life’ [26]. However, this is not specific to chronic progressive neurological diseases. Although it has been addressed as care in chronic neurological disease as a symposium at the Society for Neurotherapy [27], no courses on palliative care can be found among the e-learning courses of the Japanese Society of Internal Medicine and the Japanese Neurological Society. Furthermore, in May 2021, guidelines on end-of-life care (EOLC) for non-cancer diseases [28] were published, but these are not currently available to the public.

Thus, neurologists have very limited opportunities to receive training in palliative care during their pre- and post-graduate education periods, and only if they make efforts to acquire their training for themselves.

## 5. Issues Affecting Palliative Care

### 5.1. Understanding in Japan about Palliative Care

The general public and patients with cancer believe that palliative care is cancer-specific and is provided by palliative medicine specialists. Therefore, even if general palliative care is provided, they may not be aware of this care. Furthermore, in end-stage neurological diseases, the idea of palliative care is not even considered, and the need for palliative care is not perceived. Some healthcare professionals, including doctors and those at the University Hospital, have started to realize that there are palliative care needs for non-cancer diseases. Nurses, on the other hand, have long prided themselves on being palliative care providers, and they are aware that there are palliative care needs for terminal neurological diseases [18].

### 5.2. Lack of Teams

Most clinics in Japan only provide physician and nursing involvement [29] (Table 1), and multidisciplinary assessment and management and team approaches are rarely provided within the clinic. Collaboration with other professionals from the multidisciplinary team, such as speech therapists and social workers, is rarely available [30] (Table 1).

As a result of the lack of a multidisciplinary team approach, multidisciplinary case conferences are difficult to arrange, as the various professionals may not have time to attend and may be located in separate facilities, as described above. In Japan, long-term care insurance is a system that provides welfare and medical services at home and in care facilities to people who require long-term care due to age-related illnesses and who need nursing care and medical care. To use long-term care insurance, the patient must have a written opinion form from a doctor. After the meeting to determine the level of care, the user can select a care manager. The actual service starts after the service manager meeting which the care manager held [29]. In this system, 53.7% of the opinion forms are filled at clinics [31], when the patients, who require long-term care insurance, attend. The service manager meeting [32] is considered extremely important for achieving multidisciplinary assessment and management, a team approach to care, and patient- and family-centered care, but only 7.8% of doctors are able to attend this meeting [33]. Thus, case conferences, which should include the service manager, are rarely possible in neurology clinics.

Moreover, in hospital wards, when a team approach is used, the same team deals with many patients, and they are able to gain experience as a team. In contrast, in clinics, teams are organized for each patient, and different patients have different teams, making it difficult to accumulate this experience as a team.

## 6. Proposals for Neurology Clinics in Japan to Become Palliative Care-Ready

These issues do affect the provision of effective palliative care for people with neurological disease. Other studies have shown that there are barriers to developing the palliative care approach. Gofton et al. identified various barriers to providing palliative care for end-stage neurological disease [19]. They identified three key challenges affect palliative care in neurology: (1) uncertainty with respect to prognosis, support availability, and disease trajectory, (2) inconsistency in information, attitudes and skills among care providers, care teams, caregivers and families, and (3) existential distress specific to neurological disease, including emotional, psychological, and spiritual distress resulting from loss of function, autonomy, and death. A further study also pointed to a lack of education, including communication skills, and nine recommendations were made including (1) development of training in simulated patient meetings for trainees on dealing with emotions, communicating bad news, and setting goals of care; (2) the consideration of palliative care aspects, such as symptom management and caregiver issues and when conducting case conferences on progressive neurological disease; (3) neurology residents should have resources for debriefing emotionally challenging end-of-life cases; (4) identification of a palliative care champion within the neurology faculty to work with the palliative care faculty to develop education for neurology residents; (5) provision of inpatient palliative care rotations for neurology residents; (6) provision of palliative care education to neurologists, particularly in undertaking family meetings; (7) provision of education to faculty to improve their teaching skills, including how to teach interactively and provide feedback to residents; (8) teaching about family meetings as one procedure, taking time to do so, as well as reporting and giving feedback; (9) emphasis of the importance of palliative care in neurology, with assessment of resident’s competence in these areas, as part of the skills and achievement assessment [18].

From this background it may be possible to make recommendations for the education and practice of palliative care in neurology clinics in Japan, as described below.

### 6.1. Palliative Care Education

The most important issue is how to ensure that physicians in neurology clinics are provided the opportunity for education, as often these physicians will be the main providers of basic palliative care. It may be necessary to provide physicians in neurology clinics with incentives and recognition of general palliative care to improve the effectiveness of the care. For instance, in Japan, doctors are required to undergo training through e-learning before they can prescribe benzodiazepines for more than one year [34]. A similar training requirement could be included in the insurance scoring of general palliative care to encourage neurologists to provide more effective care. This could dramatically increase the practice of general palliative care, but to achieve this goal, the government needs to be encouraged by many authorities, including the Japanese Society of Palliative Medicine and Japanese Society of Neurology.

### 6.2. Primary and Secondary General Palliative Care

To enable multidisciplinary assessment and management with a team approach in neurology clinics, it may be necessary to provide primary general palliative care within the clinic by doctors and nurses only. This would then be supported by secondary general palliative care, which is provided in collaboration with multiple professionals in the community. Primary general palliative care would require changes within clinics, but with the determination and ingenuity of doctors in clinics who have time constraints both during and after clinic hours, they may be able to allow sufficient time for conferences with nurses. Within secondary general palliative care, it is also necessary to devise means of collaboration, such as telephone, fax, and web conferencing, to reduce the time and space constraints of doctors in clinics as much as possible. In this way, both primary and secondary general palliative care techniques are considered to be ways to ensure effective general palliative care in neurology clinics.

### 6.3. Lack of Experience of the Team

To facilitate care when a team approach is used in neurology clinics, a mechanism for sharing the experience of each team in the community would be necessary, to allow continuity of care of the patient and family. The traditional approach would be to conduct training sessions and discuss case studies. However, even if a workshop is conducted, doctors in clinics who cannot even attend the service manager meeting may not be able to participate in further activities. It may be necessary to encourage doctors in neurology clinics to actively participate or to devise a way to conduct the workshops with less time and space barriers to their participation, such as consideration of online workshops or on-demand delivery.

It may also be possible to gather and exchange the experiences of teams on a common portal site on the web. However, this requires careful consideration of the confidentiality of personal information and consideration who should bear the costs of setting up, managing, and maintaining the portal site. These solutions appear to be methods that can be adopted to address the inexperience of teams in neurology clinics.

## 7. Summary

This paper describes the problems in providing palliative care at neurology clinics in Japan and proposes solutions to improve the care. Although this review applies to Japan, the issues of identifying problems and proposing solutions may be relevant to other countries, dependent on the differences healthcare systems. The need for palliative care in neurology clinics in all countries is likely to increase, and the issues and suggestions presented here may provide helpful insights to everyone with the aim of improving care for patients and their families.

## Figures and Tables

**Table 1 brainsci-12-01707-t001:** Average numbers of professionals per 100,000 population, hospital, and clinic in Japan.

Professionals	Dr	Ns	PT	OT	ST	SW
Per 10^5^ population	278.7	744.8	72.37	37.77	13.13	11.28
Per hospital	30.03	111.2	10.83	6.233	2.178	1.789
Per Clinic	1.336	1.360	0.131	0.026	0.008	0.013

Dr, average number of doctors; Ns, average number of nurses; PT, average number of physiotherapists; OT, average number of occupational therapists; ST, average number of speech therapists; SW, average number of social workers.

## Data Availability

Not applicable.

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
