# Peer review of "A Narrative Review of Problems in Learning and Practicing Palliative Care in Neurology Clinics in Japan and Proposed Solutions"

_brainsci, 2022, doi:10.3390/brainsci12121707_

Round 1

Reviewer 1 Report

The topic area is interesting but I think the article would have worked better if case examples had been used together with evidence of systems/solutions that have worked in other areas or in other specialties to apply to pall care skills for the neurologist. In my opinion too much opinion and not enough evidence to back the solutions. Not having ever worked in neurology or palliative care in Japan, I struggled a little to follow the set-up and what the author meant by some terms e.g. primary basic palliative care. 

Author Response

Thank you for peer review. As you mentioned, examples of what works in outpatient clinics were missing, so in the introduction we cited examples from USA and Czechoslovakia in Parkinson's disease and related disorders. We have also carefully examined the content and tried to keep the text and content as compact and focused on the main points as possible.

Reviewer 2 Report

The introduction could be improved further by outlining the evidence of the benefit for patients with chronic progressive neurological conditions and their families of a palliative care approach and training in general palliative care.  Explain why training to identify and support palliative care needs will improve care.  Improvements to care and patient experience should be the driver for system change.

We tend to consider palliative care as either general palliative care or specialist palliative care and I would suggest replacing the terms basic and specialised which are referred to in the article.

The article summarises some of the barriers to MDT working; virtual MDTs utilising videoconferencing is likely to be a very useful tool and would enable shared learning/complex case discussion between teams (hospital, clinics and community) and learning from experience.  

Good communication underpins palliative care and so at the very least training and development in this core skill could be a first step to improving care - this wasn't identified in the article.

It would be good to conclude with 1 or 2 solutions that could most easily be implemented - a take home message.

Author Response

Thank you for your peer review. As you mentioned, there were few articles that proved that the palliative care approach is effective for end of life care and also useful for neurological diseases, so we have supplemented the list with some articles that provide evidence.
In addition, basic and secondary palliative care has been corrected to general palliative care and specialist palliative care.
Also, communication problem was not mentioned, but these have been indicated before the proposal.

Reviewer 3 Report

This paper can be published in a more local journal.

A more academic language should be used. The author always says 'I'.

It does not add an innovation to the literature.

Author Response

Thank you for the peer review. As you pointed out, I should probably change the first person to We so that this opinion is not taken personally, but as this is a paper with only one author, the first person is I, I used. Although you pointed out the lack of innovation, we are submitting this paper to let you know the current situation in palliative care in neurology clinics. Palliative care in neurology clinics is likely to become increasingly necessary in the future, but there are no papers from neurology clinics. I cannot deny that I do not have sufficient evidence and consideration, but I believe that the mere publication of such an article from a neurology clinic is innovative enough.

Reviewer 4 Report

Interesting topic demonstrates the author's will to improve existing practices/policies to provide better care.

It is a reflection whose context is particular, although the author has tried to contextualize it with more macro/country data.

In my opinion, the suggestions presented need to be discussed by peers, for example, using a Delphi/focus group to validate that the proposed recommendations are appropriate and listening to other perspectives and ideas. Otherwise, without neglecting the suggestions presented, it is just a perspective/vision and not the concept of various experts/guidelines existing in other contexts that can be transferred to the context under study.

Author Response

Thank you for your peer review. As you have pointed out, perhaps we should also consolidate the opinions of other doctors in the same situation and brush up on this so that it does not become a self-serving opinion. However, at least within 10 days of receiving your opinion, we do not have the time and the ability to consolidate such opinions, so I am sorry that I have not been able to go that far in my reply to the peer review this time.
Incidentally, I have been organising study groups with doctors, nurses, rehabilitation professionals and care professionals involved in the medical treatment of neurological diseases in the community once or twice/year since 2018. If you want to gather opinions, it might be a good idea to ask the members of that group for their opinions.

Round 2

Reviewer 1 Report

Despite a response and changes form the author I'm afraid my overall view is unchanged. 

Reviewer 4 Report

The document has improved. A much more clearer picture of the reality. I still have the opinion that it seems just a point of view and it is not a global opinion formed by the available evidence. However it is a more robust document.